# Numerical Study on the Fluid Flow and Heat Transfer Characteristics of Al_2_O_3_-Water Nanofluids in Microchannels of Different Aspect Ratio

**DOI:** 10.3390/mi12080868

**Published:** 2021-07-24

**Authors:** Huajie Wu, Shanwen Zhang

**Affiliations:** 1College of Traffic Engineering, Yangzhou Polytechnic Institute, Yangzhou 225127, China; 2College of Mechanical Engineering, Yangzhou University, Yangzhou 225127, China; swzhang@yzu.edu.cn

**Keywords:** microchannel, nanofluid, heat transfer enhancement, numerical simulation

## Abstract

The study of the influence of the nanoparticle volume fraction and aspect ratio of microchannels on the fluid flow and heat transfer characteristics of nanofluids in microchannels is important in the optimal design of heat dissipation systems with high heat flux. In this work, the computational fluid dynamics method was adopted to simulate the flow and heat transfer characteristics of two types of water-Al_2_O_3_ nanofluids with two different volume fractions and five types of microchannel heat sinks with different aspect ratios. Results showed that increasing the nanoparticle volume fraction reduced the average temperature of the heat transfer interface and thereby improved the heat transfer capacity of the nanofluids. Meanwhile, the increase of the nanoparticle volume fraction led to a considerable increase in the pumping power of the system. Increasing the aspect ratio of the microchannel effectively improved the heat transfer capacity of the heat sink. Moreover, increasing the aspect ratio effectively reduced the average temperature of the heating surface of the heat sink without significantly increasing the flow resistance loss. When the aspect ratio exceeded 30, the heat transfer coefficient did not increase with the increase of the aspect ratio. The results of this work may offer guiding significance for the optimal design of high heat flux microchannel heat sinks.

## 1. Introduction

With the continuous miniaturization and integration of electronic devices, the heat flux density continues to increase, and conventional cooling methods are no longer effective. Therefore, the heat dissipation problem of high heat flux density has been a research hotspot in the field of heat transfer [1,2]. In 1981, Tuckerman and Pease [3] proposed a silicon microchannel heat sink, which uses deionized water as a liquid cooling medium and whose heat dissipation capacity can be as high as 790 W/cm^2^. In 1995, Choi and Eastman [4] proposed the concept of nanofluids; a proportion of solid particles with diameters less than 100 nm were added into a base fluid with low thermal conductivity, and the resulting suspension with high thermal conductivity was found to be relatively stable. This special liquid can significantly improve the convective heat transfer capacity of cooling media. The addition of nanoparticles to high-Prandtl number liquids significantly increases the heat transfer performance of micro heat-sinks [5]. An increase in nanoparticle concentration can lead to an increase in thermal conductivity and viscosity and an increase in nanoparticle size [6]. Nanofluids contribute to the improvement of heat transfer processes and reduce and optimize thermal systems. Different properties, such as wettability and thermal conductivity, can be adjusted by altering the nanoparticles’ concentration, thereby making nanofluids suitable for a wide range of applications [7]. Nanofluids contain metal or nonmetal particles with nanometer sizes and exhibit much greater thermal conductivity. M. Goodarzi’s [8] expression for calculating the enhanced thermal conductivity of nanofluids has been derived from the general solution of the heat conduction equation in spherical coordinates and the equivalent hard-sphere fluid model representing the microstructure of particle/liquid mixtures. The cooling method of microchannel heat sinks combined with nanofluids has become one of the effective ways to solve the heat dissipation problem of high heat flux. By comparing the heat transfer characteristics of trapezoidal, semicircular and rectangular cross-section microchannels, Vinoth et al. [9] found that, compared with rectangular and semicircular cross-section microchannels, trapezoidal cross-section microchannels have the best heat transfer effect because of their larger wall area and effective inlet length, but a larger pressure drop. For some special-shaped cross-sections, Alfaryjat et al. [10] used numerical simulation methods to study hexagonal, circular and rhombic cross-section microchannels. The results show that the heat transfer coefficients of hexagonal cross-section microchannels are the highest, followed by circular and rhombic cross-section microchannels. Ahmed et al. [11] used the three-dimensional numerical simulation method to optimize the microchannel with triangular, trapezoidal and rectangular grooves. The results showed that the optimized microchannel with trapezoidal groove had the best heat transfer effect, the Nusselt number increased by 51.59% and the friction coefficient increased by 2.35% compared with the optimization. Kumar [12] used the finite volume method to simulate and optimize the trapezoidal microchannel. The results show that the heat transfer performance of trapezoidal microchannels with semicircular grooves is 16% higher than that of trapezoidal microchannels with a rectangular groove, but the friction coefficient is 18%. At present, the structure optimization of microchannels has some limitations; namely, the processing is very difficult, and the cost is high. Researchers gradually try to improve the heat transfer by changing the flow medium in the microchannel. Farsad et al. [13] numerically simulated the heat transfer performance of Al_2_O_3_, CuO and Cu-H_2_O nanofluids in copper rectangular microchannels. The results show that the thermal conductivity of metal nanofluids is higher than that of metal oxide nanofluids. Researchers have conducted numerous studies to obtain the optimized aspect and states in order to improve the heat transfer of equipment and use various nanofluids. Shi Xiaojun et al. [14] carried out a multi-objective optimization design on a single-layer nanofluid rectangular microchannel. The results show that the pump power and thermal resistance are more sensitive to the channel width and spacing ratio than the aspect ratio. Naphon and Khonseur [15] used air as a cooling medium to conduct an experimental study on the flow and heat transfer characteristics of microchannels with different heights and widths in the Reynolds number range of 200–1000. The results showed that the height and width of rectangular microchannels exert a significant impact on their heat exchange effect and resistance loss. Studies of this research indicate that the fluid in the indented sections has a higher heat transfer with the heated wall. Karimipour et al. [16] numerically studied a two-dimensional indented rectangular microchannel. They concluded that by increasing the volume fraction of nanoparticles, the thermal efficiency of the nanofluid is enhanced. Yari Ghale et al. [17] numerically studied the laminar and forced flow of a Water/Al_2_O_3_ nanofluid in an indented microchannel by using two-phase or single-phase methods. Their results showed that the Nusselt numbers and friction factors in an indented microchannel are higher compared to the smooth microchannel, and therefore, these parameters can improve fluid flow efficiency by increasing the width of the rib. A segmental analysis pertaining to the heat exchanger takes place to evaluate the influence of nanofluid usage on the heat transfer coefficient, the exchanger’s length and its pressure drop. When the volume fraction of Al_2_O_3_ nanofluids is 5%, the heat transfer coefficient is increased by 10% compared with pure water, and the pressure drop is significantly reduced [18]. T Raghuraman [19] used pure water as a cooling medium to study the effect of the microchannel aspect ratio on heat transfer performance; the study showed that the microchannel aspect ratio influences the heat transfer coefficient, pumping power, pressure drop and heat transfer performance at different Reynolds numbers. A large aspect ratio can enhance heat transfer, but it can also increase power consumption. Based on the computational fluid dynamics (CFD) method, Mohamadpour et al. [20] carried out a numerical simulation study on the heat transfer efficiency of the cooperative jet in the microchannel. It was found that increasing the jet frequency and pulse amplitude can significantly improve the heat transfer ability of the microchannel.

In the current work, a rectangular microchannel heat sink is used as the research object, and a three-dimensional flow and heat transfer numerical simulation study is conducted on the basis of the computational fluid dynamics method. Water/Al_2_O_3_ nanofluids of different concentrations are utilized as the cooling medium, and their influence on the heat transfer performance of microchannels is analyzed. This work also focuses on the evaluation of the flow resistance characteristics and heat transfer laws of microchannels with different aspect ratios. The purpose of the study is to provide theoretical guidance for the optimal design of high heat flux density microchannel heat sinks.

## 2. Numerical Method and Model Description

### 2.1. Mathematical Model

The single-phase fluid model is commonly used to study the flow and heat transfer of nanofluids in microchannels. The nanoparticles in nanofluids are considered to be uniformly distributed in base fluids, and they are in thermal equilibrium. No relative slip velocity exists between nanoparticles and base liquids, and the flow is regarded as an incompressible steady laminar flow.

On the basis of these assumptions, the governing equations of nanofluid flow and heat transfer can be expressed as:(1)$$\frac{\partial }{\partial x_{j}}\left(\rho_{\mathrm{nf}}u_{j}\right)=0$$
(2)$$\frac{\partial \left(\rho_{\mathrm{nf}}u_{i}u_{j}\right)}{\partial x_{j}}=-\frac{\partial p}{\partial x_{i}}+\frac{\partial }{\partial x_{j}}\left[\mu_{\mathrm{nf}}\left(\frac{\partial u_{i}}{\partial x_{j}}+\frac{\partial u_{j}}{\partial x_{i}}\right)\right]$$
(3)$$\frac{\partial }{\partial x_{j}}\left(\rho_{f}C_{p}{}_{f}Tu_{j}\right)=\frac{\partial }{\partial x_{j}}\left[\lambda_{\mathrm{nf}}\frac{\partial T}{\partial x_{j}}\right]$$
where *u*_i_ and *u*_j_ are velocity components, *x*_i_ and *x*_j_ are Cartesian coordinate components, *p* is the pressure in the flow field, *T* is the temperature, *ρ*_nf_ is the density of nanofluids, and *μ*_nf_ is the dynamic viscosity. *C*_pf_ is the specific heat capacity, and *λ*_nf_ is the thermal conductivity. The calculation formula is as follows [21,22]:(4)$$\rho_{\mathrm{nf}}=\left(1-\alpha \right)\rho_{w}+\alpha \rho_{p}$$
(5)$$\mu_{\mathrm{nf}}=\left(1+0.025\alpha +0.015\alpha^{2}\right)\mu_{w}$$
(6)$$C_{\mathrm{pf}}=\left[\left(1-\alpha \right){\left(\rho C_{p}\right)}_{w}+\alpha {\left(\rho C_{p}\right)}_{p}\right]/\rho_{\mathrm{nf}}$$
(7)$$\lambda_{\mathrm{nf}}=\frac{\lambda_{p}+(n-1)\lambda_{w}-(n-1)\alpha (\lambda_{w}-\lambda_{p})}{\lambda_{p}+(n-1)\lambda_{w}+\alpha (\lambda_{w}-\lambda_{p})}\lambda_{p}$$

The subscripts w and p denote the corresponding thermophysical properties of the base fluid and nanoparticles, respectively. *α* represents the volume fraction of nanoparticles, and n is the shape factor of nanoparticles. In this work, nanoparticles are regarded as regular spheres with a value of *n* = 3.

The heat distribution in the solid area of the heat sink can be calculated by the following formula, *T_s_* is the temperature of the solid region; *λ_s_* is the thermal conductivity of the solid region:(8)$$\frac{\partial }{\partial x_{i}}\left(\lambda_{s}\frac{\partial T_{s}}{\partial x_{i}}\right)=0$$

### 2.2. D Model and Boundary Conditions

A microchannel heat sink is usually composed of more than 10, or even dozens, of microchannels, and complete modeling and simulation require a large amount of computing resources. As a result of the symmetry of the model, a typical microchannel heat sink unit can be extracted for simulation. The structure and size of the microchannel heat sink used in this study are shown in Figure 1. The width *W* of the heat sink unit is 1 mm, the length *L* is 50 mm, and the microchannel width *W*_c_ is 0.5 mm. The microchannel height *W*_h_ values are 5, 10, 15, 20 and 25 mm; they correspond to five different aspect ratios, that is, *HW = W*_h_*/W*_c_ with values of 10, 20, 30, 40 and 50, respectively. The bottom height of the heat sink is set at 6 mm. Symmetrical boundary conditions are set on both sides of the heat sink, and a uniform heat flux *q* = 0.8 MW/m^2^ is set at the bottom. The top wall is set as the adiabatic boundary. The inlet of the microchannel fluid adopts the velocity inlet boundary, the inlet fluid temperature is set at 300 K, the outlet of the microchannel fluid is set as pressure outlet, the upper wall of the fluid domain adopts an adiabatic solid wall boundary condition and the interface between fluid and solid adopts a coupled heat flow boundary condition. In this paper, ICEM CFD software is used for 3D modeling and mesh generation, and the general CFD software FLUENT 15.0 is used for numerical simulation. According to the geometric models with different aspect ratios, three sets of grids are divided to analyze the grid independence under the maximum Reynolds number.

### 2.3. Model Validation

The experimental data of Lei [23] are used for comparison to verify the prediction performance of the mathematical model. The microchannel width *W*_c_ is 0.1 mm, the height *W*_h_ is 0.5 mm, the length *L* is 10 mm, and the heat flux *q* at the bottom of the heat sink is 0.6 MW/m^2^. The cooling medium is pure water. In the calculation, the volume fraction of the nanoparticles α is 0.

The characteristic scale of microchannels can be defined as:(9)$$D_{h}=\frac{2W_{h}W_{c}}{W_{h}+W_{c}}$$

The microchannel Reynolds number is:(10)$$Re=\frac{\rho_{f}U_{\mathrm{in}}D_{h}}{\mu_{f}}$$
where *U*_in_ is the fluid inlet velocity.

The average temperature *T*_c_ of the heat transfer surface and the average temperature *T*_f_ of the whole microchannel fluid can be respectively defined as:(11)$$T_{c}=\frac{\int TdA}{\int dA}$$
(12)$$T_{f}=\frac{\int \rho_{f}TdV}{\int \rho_{f}dV}$$

Therefore, the average heat transfer coefficient can be expressed as:(13)$$h=\frac{qA_{b}}{A_{c}\left(T_{c}-T_{f}\right)}$$
where *A*_b_ is the heating area of the bottom of the thermal sink and *A*_c_ is the heat exchange area between the fluid domain and the solid domain in the microchannel.

The average Nusselt number is defined as follows:(14)$$Nu=\frac{hD_{h}}{\lambda_{f}}$$

The comparison between the average Nusselt number predicted by the mathematical model and the experimental results is shown in Figure 2. The results show that the proposed model achieves certain accuracy and reliability in the prediction of fluid laminar flow and heat transfer in microchannels. The article only carried out simulation research, and there will be errors between the simulation results and the experiment. There are many sources of error, such as the flow may not be completely laminar, the number of grids, the finite element method and so on.

## 3. Result Analysis and Discussion

### 3.1. Influence of Nanoparticle Volume Fraction

Take the microchannel heat sink with aspect ratio *HW* = 10 (*HW* = *W*_h_/*W*_c_) as an example. The three-dimensional flow and heat transfer of the nanofluids are simulated in the Reynolds number range of 100–500, and the nanoparticle volume fractions α are 0.5% and 5%, respectively. According to Formula (10), the Reynolds number is directly related to density, inlet velocity, characteristic scale and viscosity, while density and viscosity are related to the content of solid particles in nanofluids. In this paper, the physical properties of the fluid and the characteristic scale of the channel are determined by giving the solid particle content and aspect ratio in the nanofluid. Finally, the Reynolds number can be changed by adjusting the inlet velocity. The temperature distributions in the middle section of the heat sink are extracted for comparison, and the results are shown in Figure 3. The laws of the temperature distributions in the middle sections of two types of nanofluids at different Reynolds numbers are similar. The temperature at the bottom of the heat sink is the highest, and the temperature in the solid region decreases gradually along the height of the microchannel. The temperature of the fluid at the center of the fluid domain of the microchannel is relatively low. At the fluid structure coupling heat transfer surface, the fluid temperature is relatively high because of the heating of the solid surface. At the same time, with the increase of the Reynolds number, the heat sink temperature and fluid temperature decrease significantly. When the Reynolds number remains the same, the heat sink temperature and fluid temperature of the nanofluid with a nanoparticle volume fraction of 5% are significantly lower than those of the nanofluid with a nanoparticle volume fraction of 0.5%.

The average temperature on the heat exchange surface of the fluid and solid domains under each working condition is obtained. The results are shown in Figure 4. With the increase of the Reynolds number, the average temperature on the heat transfer surface of the two types of nanofluids decreases. At the same Reynolds number, increasing the volume fraction of nanoparticles in the nanofluid can reduce the temperature on the heat transfer surface. In the range of Reynolds numbers studied in this work, the average temperature difference on the heat transfer surface caused by the differences in nanoparticle volume fraction decreases with the increase of the Reynolds number. When the Re is 100, the average temperature difference between the nanofluid with a nanoparticle volume fraction of 5% and that with a nanoparticle volume fraction of 0.5% is 6.3 K. When the Re is 500, the average temperature difference of the heat exchange surface decreases to 2.6 K. This result shows that the average temperature of the heat exchange surface can be reduced by increasing the nanoparticle volume fraction under the condition with a low Reynolds number.

Figure 5 shows the effects of the nanoparticle volume fraction on heat transfer coefficients with different Reynolds numbers. The results show that the average heat transfer coefficient increases with the increase of the Reynolds number and that the increase of the nanoparticle volume fraction can improve the heat transfer ability of the nanofluids. As shown in Figure 6, in order to comprehensively consider the difference of fluid thermal conductivity caused by different volume fractions of nanoparticles, the Nusselt number is used as the evaluation index for comparative analysis, and the same conclusion can be obtained.

Figure 7 shows the distribution of the pressure difference between the inlet and outlet of the microchannel with different volume fractions. The drag loss of nanofluids through microchannels increases with the increase of the Reynolds number. However, the drag loss of nanofluids with a nanoparticle volume fraction of 5% is greater than that of nanofluids with a nanoparticle volume fraction of 0.5%. At the same time, because the volume fraction of nanoparticles affects the density and dynamic viscosity of nanofluids, the velocity of the microchannel inlet must be adjusted to keep the same Reynolds number. As shown in Figure 8, with the increase of the Reynolds number, the inlet velocity of the 5% nanofluid is greater than that of the 0.5% nanofluid. Pumping power is introduced as the evaluation index to reasonably evaluate the synergistic effect of inlet velocity and resistance loss. Its physical meaning is the external work required for nanofluids to pass through microchannels. Pumping power p is expressed as:(15)$$P=N\cdot U_{\mathrm{in}}\cdot W_{h}\cdot W_{c}\cdot_{\Delta }p$$

*N* is the number of microchannels in the whole heat sink, and the value is *N* = 1.

The variation of pumping power with the Reynolds number is shown in Figure 9. With the increase of the Reynolds number, the pumping power of the nanofluid with a nanoparticle volume fraction of 5% is significantly higher than that of the nanofluid with a nanoparticle volume fraction of 0.5%. This result shows that the heat transfer performance cannot be improved by increasing the nanoparticle volume fraction because doing so greatly increases the power consumption of the whole system. At the same time, a high volume fraction renders the nanoparticles in nanofluids unable to maintain a stable and uniform suspension state [23]. Therefore, in engineering applications, the nanoparticle volume fraction in nanofluids needs to be maintained at a low level.

### 3.2. Influence of the Aspect Ratio of Microchannels

This work studies nanofluids with a nanoparticle volume fraction of 5%. The different aspect ratios (*HW*) of microchannels can be obtained by changing their height (*W*_h_). The aspect ratios of the five microchannel heat sink models are 10, 20, 30, 40 and 50. A three-dimensional simulation of flow and heat transfer under different Reynolds numbers is conducted, and the flow and heat transfer characteristics of nanofluid microchannels with different aspect ratios are analyzed and compared. As shown in Figure 10, the pressure difference between the inlet and the outlet increases with the increase of the Reynolds number under different aspect ratios of the microchannel heat sink. The pressure difference decreases with the increase of the aspect ratio at the same Reynolds number. When the aspect ratio is 20–50, the pressure difference is not obvious. This result shows that the increase of the aspect ratio of the microchannel does not greatly enhance the flow resistance loss of nanofluids under the parameters studied in this work. Figure 11 shows the variation of the resistance coefficient f of the microchannel with the Reynolds number. The simulation results of the five different aspect ratios show that the drag coefficient decreases with the increase of the Reynolds number. Moreover, the differences in the drag coefficients caused by different aspect ratios decrease with the increase of the Reynolds number.

Figure 12 shows the variation of the average temperature at the bottom of the microchannel heat sink with the Reynolds numbers at different aspect ratios. The results show that the mean bottom temperature, corresponding to the five microchannel heat sinks, decreases with the increase of the Reynolds number and that increasing the aspect ratio of the microchannel can reduce the bottom temperature of the heat sink. However, when the aspect ratio exceeds 20, the decrease of the heat sink’s bottom temperature drops, caused by an increase in the aspect ratio of the microchannel. Furthermore, the average temperature values at the bottom of the three microchannels with aspect ratios 30, 40, and 50 almost coincide.

The average temperature distribution of the fluid-solid surface relative to the heat transfer in the microchannels with different aspect ratios is shown in Figure 13. With the increase of the Reynolds number, the average temperature of the heat transfer surface decreases. When the aspect ratio *HW* increases from 10 to 20, the corresponding temperature drop is 16.8 K at an Re of 100. When the aspect ratio *HW* increases from 20 to 30, the corresponding temperature drop is 5.7 K. When the aspect ratio *HW* increases from 30 to 40, the corresponding temperature drop decreases to 2.9 K. When the aspect ratio *HW* increases from 40 to 50, the corresponding temperature drop further decreases to 1.8 K. As shown in Figure 14, when the aspect ratio *HW* increases from 10 to 30, the Nusselt number increases. When the aspect ratio further increases from 30 to 50, the Nusselt number does not increase significantly. This result indicates that the increase of the aspect ratio does not significantly improve the heat transfer performance of the microchannel heat sink in this range. The above analysis shows that the change of the aspect ratio of the microchannel affects its heat transfer performance and resistance characteristics. In this work, the comprehensive heat transfer performance index is used to quantitatively evaluate the synergistic effect. It is defined as:(16)$$\eta =\frac{\left(Nu/Nu_{0}\right)}{{\left(f/f_{0}\right)}^{1/3}}$$
where *f*_0_ is the resistance coefficient of the microchannel with aspect ratio *HW* = 10 and *Nu*_0_ is the average Nusselt number of the microchannel with aspect ratio *HW* = 10.

Figure 15 shows the variations of the comprehensive heat transfer performance parameters of the microchannel heat sinks with different aspect ratios and given different Reynolds numbers. In the range of the Reynolds numbers studied in this work, the comprehensive heat transfer performance parameters are greater than 1. The results indicate that for the microchannel heat sink with *HW* = 10, increasing the aspect ratio can improve its comprehensive heat transfer performance. When the aspect ratio is increased to 30, the comprehensive heat transfer performance of the microchannel heat sink does not continue to improve.

## 4. Conclusions

In this work, a microchannel heat sink is studied on the basis of computational fluid dynamics. The flow and heat transfer of two nanofluids with different volume fractions and five microchannel heat sinks with different aspect ratios in the Reynolds number range of 100–500 are simulated. The flow and heat transfer characteristics of the microchannel heat sinks are compared, and the optimal parameters of the aspect ratio are analyzed. The conclusions are as follows:(1)Increasing the volume fraction of nanoparticles can effectively reduce the average temperature of the heat transfer surface and improve the heat transfer capability of nanofluids. However, because of the dual increase of the inlet velocity and flow resistance, the power consumption of the whole system increases greatly;(2)Increasing the aspect ratio of the microchannel does not cause significant flow resistance loss, and the resistance coefficient of the microchannel tends to be consistent with the increase of the Reynolds number at different aspect ratios;(3)Increasing the aspect ratio of the microchannel can reduce the temperature of the heat sink. When the aspect ratio exceeds 30, the average temperature at the bottom of the microchannel does not decrease, and the heat transfer coefficient does not increase;(4)In the range of the parameters studied in this paper, the aspect ratio of the microchannel heat sink with a thickness of 6 mm has an optimal value. Based on the comprehensive heat transfer performance parameters, the optimal value of the aspect ratio of the microchannel heat sink is 30.

This study shows that the aspect ratio of the heat sink has a significant impact on the heat transfer performance of a microchannel, and there is an optimal value in the range of Reynolds numbers under the condition of a given thickness. Further research will be carried out for different thicknesses of heat sinks under different Reynolds number conditions to obtain a universal empirical formula for guiding engineering practice.

## Figures and Tables

**Figure 1 micromachines-12-00868-f001:**
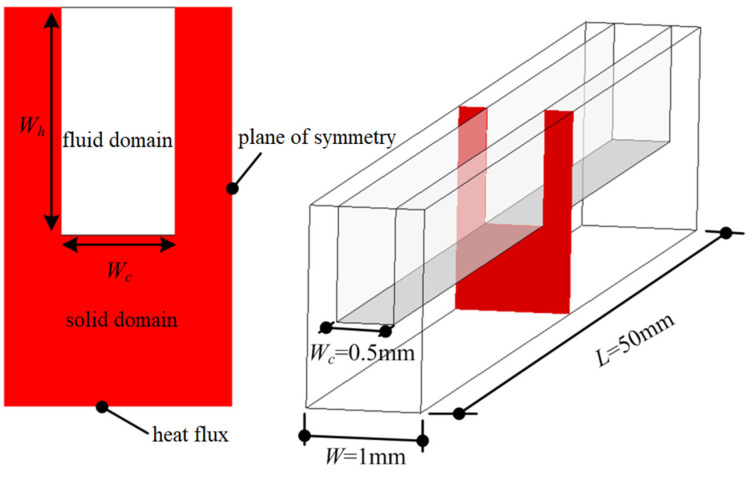
Schematic of heat sink with square cross-section.

**Figure 2 micromachines-12-00868-f002:**
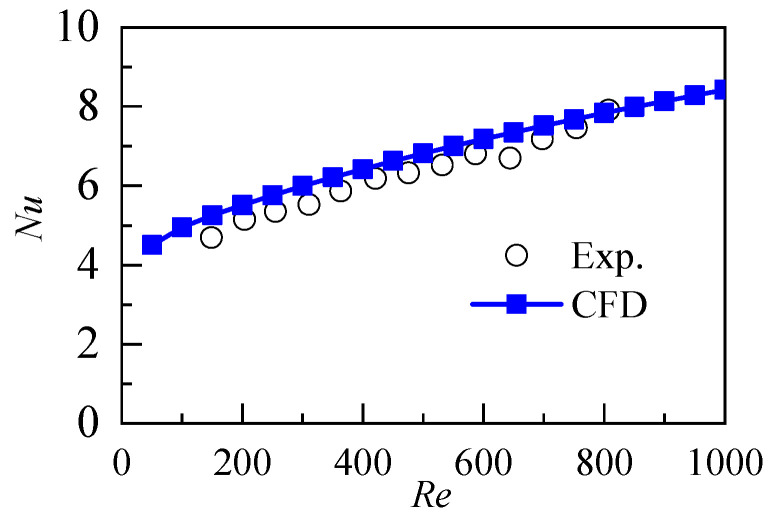
Model validation by comparing the present results with Lei et al. experiments [23].

**Figure 3 micromachines-12-00868-f003:**
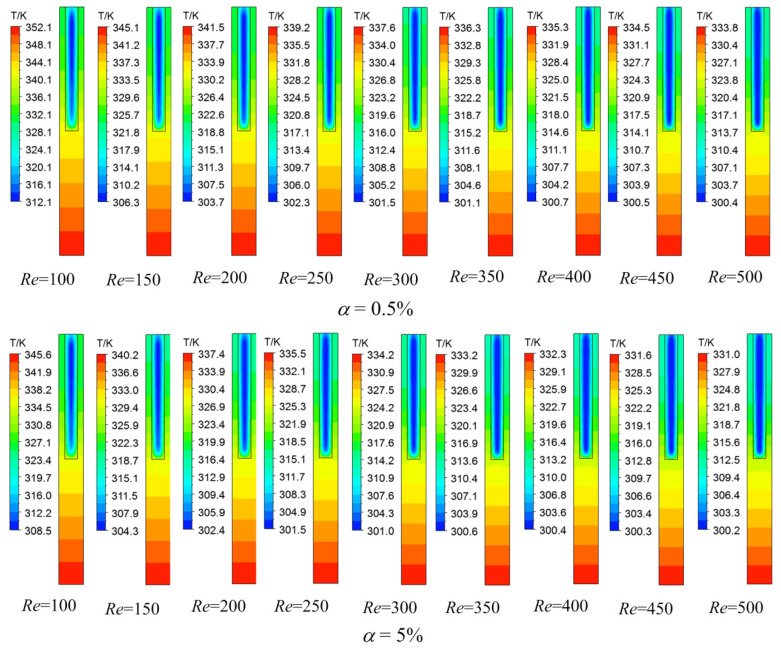
Temperature distributions of fluid and solid domains at the middle plane.

**Figure 4 micromachines-12-00868-f004:**
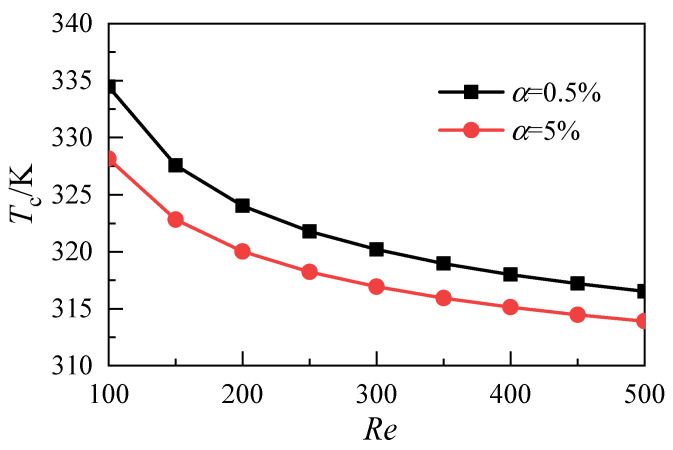
Average temperature distributions of the fluid-solid surface.

**Figure 5 micromachines-12-00868-f005:**
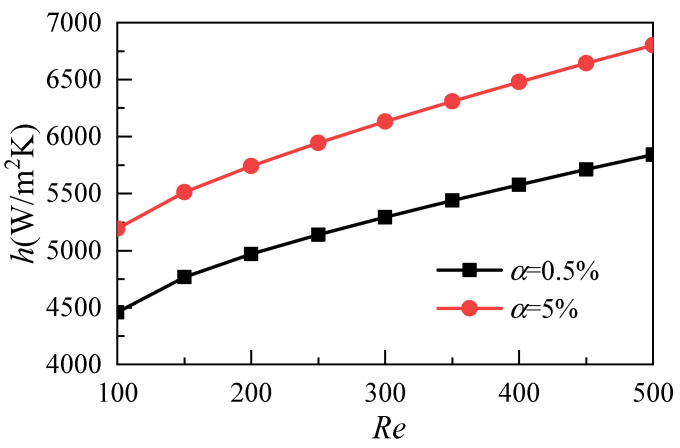
Effects of nanoparticle volume fraction on heat transfer coefficients.

**Figure 6 micromachines-12-00868-f006:**
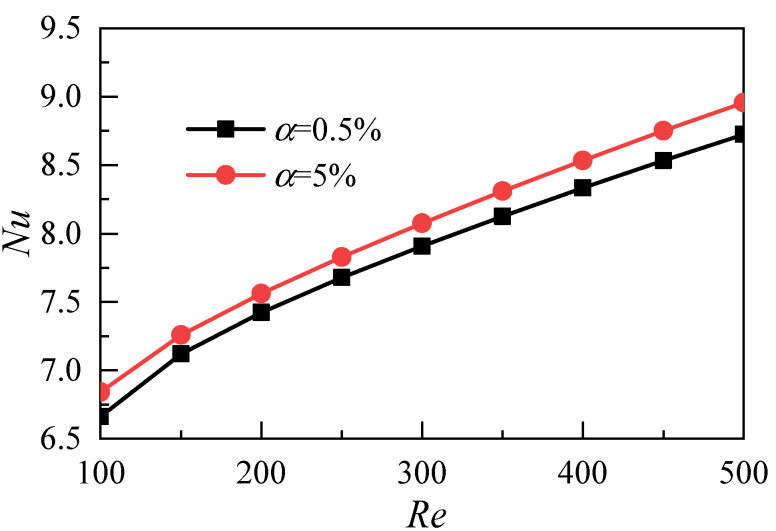
Influence of nanoparticle volume fraction on Nusselt number.

**Figure 7 micromachines-12-00868-f007:**
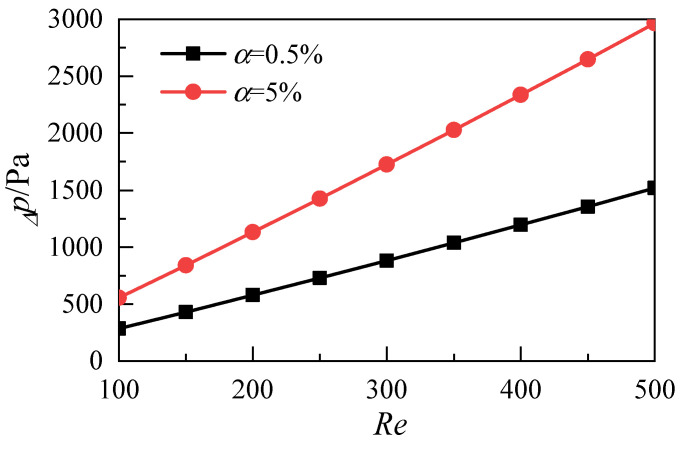
Influence of nanoparticle volume fraction on pressure difference.

**Figure 8 micromachines-12-00868-f008:**
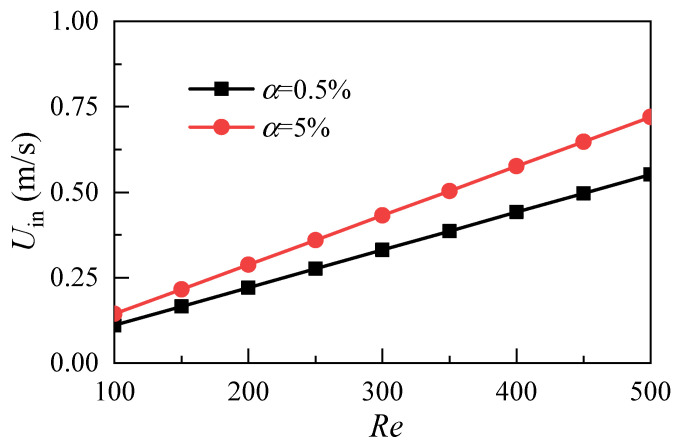
Effect of nanoparticle volume fraction on inlet velocity.

**Figure 9 micromachines-12-00868-f009:**
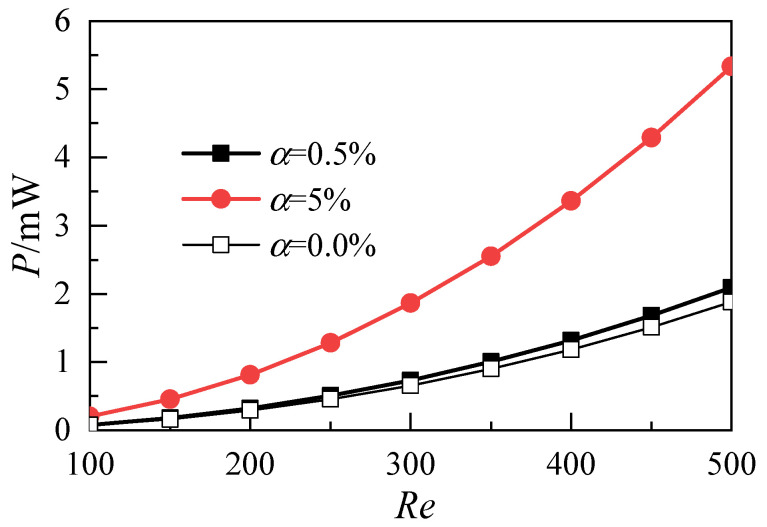
Effects of nanoparticle volume fraction on pumping power.

**Figure 10 micromachines-12-00868-f010:**
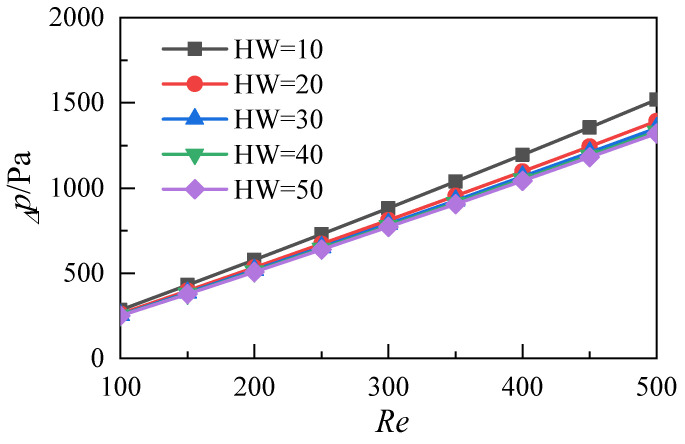
Distributions of pressure difference between the inlet and the outlet of the microchannel heat sink with different aspect ratios.

**Figure 11 micromachines-12-00868-f011:**
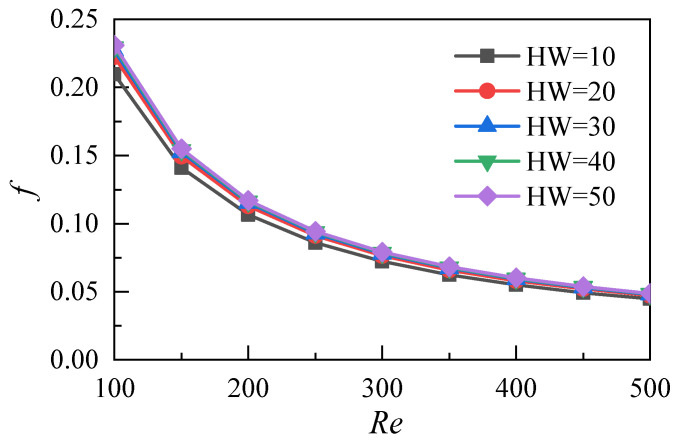
Distributions of friction coefficients.

**Figure 12 micromachines-12-00868-f012:**
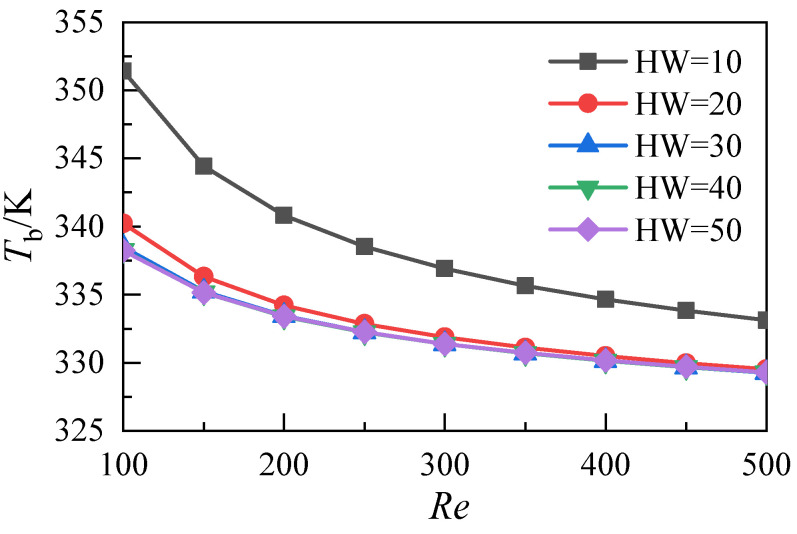
Temperature distributions of the heat sink bottom with different aspect ratios.

**Figure 13 micromachines-12-00868-f013:**
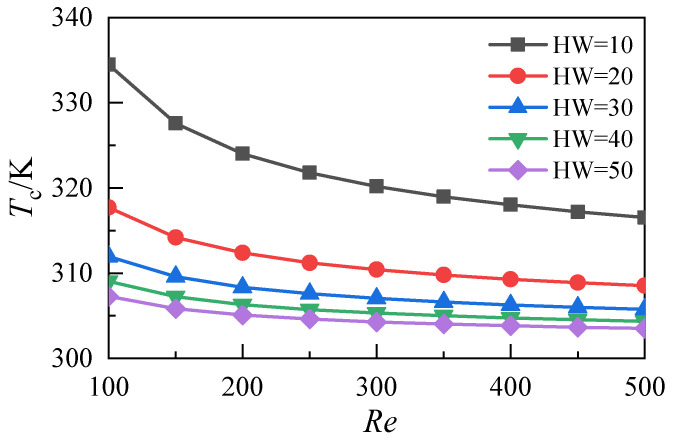
Temperature distributions of fluid-solid surface in microchannels with different aspect ratios.

**Figure 14 micromachines-12-00868-f014:**
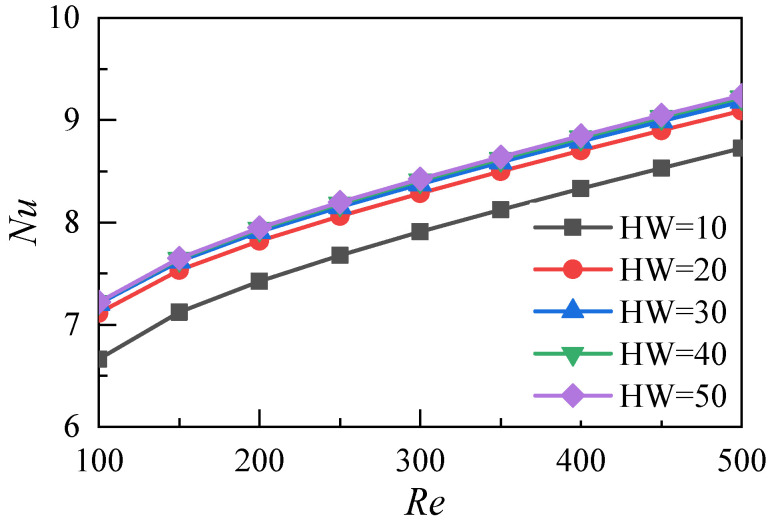
Nusselt number distributions of microchannels with different aspect ratios.

**Figure 15 micromachines-12-00868-f015:**
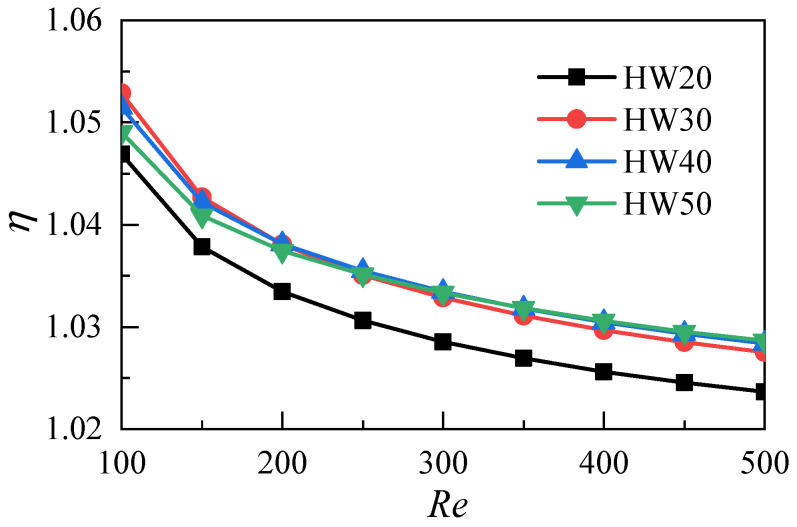
Comparisons of comprehensive heat transfer performances of microchannels with different aspect ratios.

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
