# Peer review of "Numerical Study on the Fluid Flow and Heat Transfer Characteristics of Al2O3-Water Nanofluids in Microchannels of Different Aspect Ratio"

_micromachines, 2021, doi:10.3390/mi12080868_

Round 1
Reviewer 1 Report
Forced convection results for laminar flow are shown.
- In world literature, such channels are called mini-channels;
- In paragraph 36 and 40 reference is made to several studies [5-11] andv[11-17]. It is considered that it should not include the references and make a short description of what happens in each case study;
- Error analysis is missing;
- Correlation of other authors needs to be shown;
- No discussions, no comparisons of the results with the results of other authors.
Unfortunately, the results were very poorly compiled.
Author Response
Thank you for your comments. The paper has been revised. Please refer to it.

Reviewer 2 Report
This manuscript provides insight into microchannel design optimization by using numerical modeling. The authors look at two practical design choices in nanofluid microchannel heat exchanger design, nanoparticle volume fraction, and the aspect ratio of the microchannels. Performance was assessed in terms of the pumping power, the average temperature of the heated surface, and the effective heat transfer coefficient for the heat exchanger. These are typical parameters when designing nanofluid microchannel heat exchangers, and there is a novelty in the presented approach because there is not much work considering all three design parameters simultaneously.
The manuscript's introduction provides a reasonable amount of review of the literature on nanofluid microchannel design and performance. However, the introduction could be improved by expanding the review of other attempts at optimizing microchannel design in the literature. A quick literature search for "nanofluid microchannel design optimization" yielded several other related works that are well-cited and not included in the introduction.
Overall, the methodological approach is sound. The chosen simulations adequately support the conclusions, and validation of the model was presented. However, there are several places where the description of the methodology needs to be improved for reproducibility and fully understanding the assumptions and modeling decisions that were made. I do not believe that the answers to any of these questions will change the suitability of the results, but they need to be included for completeness. Before publication, the following comments should be addressed.
- Describe the software package that was used to run the simulations. Provide a citation for the software if applicable.
- Provide definitions for λs and Ts
- Provide a table with all the material constants used in the simulations for equations (1) – (8), i.e., density, thermal diffusivity, viscosity, and specific heat for all phases.
- Describe the boundary conditions for the interface of the fluid domain with the solid domain.
- Describe the temperature and velocity field boundary conditions for the top surface of the fluid domain.
- Describe the temperature boundary conditions for the top surface of the solid domain.
- Describe the height of the bottom portion of the heat exchanger.
- When describing changes in Reynold’s number, indicate which independent variable was being changed to achieve the change in Reynold’s number, i.e., inlet velocity, viscosity, or channel aspect ratio.
The results and conclusions are well presented. I have only a few comments and suggestions for improvement of these sections:
- Conclusion (4) should consider if there are any specific limitations to the presented design optimization. The optimal result may change if the thickness of the bottom part of the heat exchanger is different or if a different nanofluid is used.
- In Fig. 3, different color scales are used for each figure. This leads to the impression that all the figures are identical. Consider using a single color bar for all of the subfigures to visualize the differences between the simulations. (Also, consider using a sequential, perceptually uniform colormap such as “inferno” or “magma,” if available in your plotting software, instead of the “rainbow” colormap. There is a decent amount of literature and online discussion on colormap selection if you are interested. In short, sequential colormaps will help more clearly show the relative differences between your plots and will also help readability if your article is printed in grayscale by a reader.)
- The description of Fig. 6 (page 5, lines 157 – 158) is repetitive from Fig. 5. Please explain the importance of presenting the Nusselt number relationship in addition to the average heat transfer coefficient or reconsider if Fig. 6 is necessary.
- Please provide a citation for the claim that “high volume fraction renders the nanoparticles unable to maintain a stable and uniform suspension state” (page 5, line 179), as well as clarify what amount defines a “high volume fraction.”
Author Response
1.In this paper, ICEM CFD software is used for 3D modeling and mesh generation, and the general CFD software FLUENT 15.0 is used for numerical simulation. According to the geometric models with different aspect ratios, three sets of grids are divided to analyze the grid independence under the maximum Reynolds number.
2.The definition for λs and Ts has been added to the table.
3.The table with all the material constants used in the simulations for equations has been added to revised version manuscript
4. Describe the boundary conditions for the interface of the fluid domain with the solid domain.The interface of fluid domain and solid domain adopts heat flow coupling boundary condition, the relevant boundary conditions have been modified in this paper.
5. Describe the temperature and velocity field boundary conditions for the top surface of the fluid domain.The solid wall adiabatic boundary condition is adopted at the top of the fluid, and the velocity inlet boundary condition is adopted at the velocity field boundary. the relevant boundary conditions have been modified in this paper.
6. Describe the temperature boundary conditions for the top surface of the solid domain.The adiabatic boundary condition is applied to the top surface of solid domain, the relevant boundary conditions have been modified in this paper.
7. Describe the height of the bottom portion of the heat exchanger.
The bottom height of the heat exchanger shall be kept at 6 mm, the article has been revised.
8. According to formula 10, Reynolds number is directly related to density, inlet velocity, characteristic scale and viscosity, and the density and viscosity are related to the content of solid particles in nanofluids. Therefore, the fluid properties and channel characteristic scale are determined by giving different solid particle content and aspect ratio in nanofluids. The Reynolds number can be changed by adjusting the inlet velocity. This idea makes the later comparative analysis more clear.
9. The author revised the conclusion as follows: In the range of the parameters studied in this paper, the aspect ratio of the microchannel heat sink with a thickness of 6 mm has an optimal value. Based on the comprehensive heat transfer performance parameters, the optimal value of the aspect ratio of the microchannel heat sink is 30. In the future, the author's team will carry out research on heat sinks with different thickness, and expand the range of aspect ratio and Reynolds number, in order to obtain more universal empirical formula.
10. The author's team used CFD post software to map the data again. For the slender strip geometry and temperature distribution characteristics in this paper, the contrast can be slightly improved by using different color schemes and adjusting the number of cloud color scales, but the difference cannot be enlarged. The simulation results in this paper confirm the similarity of temperature distribution and the difference of temperature distribution, which can be directly reflected by the upper and lower limit of the legend of each small picture. In addition, in order to ensure the consistency of the post-processing settings used in each kind of cloud image results, this paper finally uses the default color scheme of CFD post software to redraw the image. The image adopts TIFF format and DPI is set to 300 to ensure the image quality.
11. As a common evaluation index of heat transfer ability, average heat transfer coefficient can directly evaluate the heat transfer ability of fluids with the same fluid properties. Because different nanoparticle fractions are considered in this paper at the same time, it will lead to all differences in the thermal conductivity of fluids, Nusselt number is used as the evaluation index for comparative analysis. The author has revised the manuscript
12. References have been added to the original text

Reviewer 3 Report
I would like to really thank the authors for the opportunity of reviewing their work. In my opinion, this study contains some interesting information and can be considered for publication, however, only after major revision. To that end, some corrections can be made in relation to different aspects of the study in order to improve its overall quality.
- First of all, the authors should carefully check the entire manuscript for possible typo and grammar mistakes. For example, the reference number usually followed the name of the author without leaving a space. See lines 28, 31, 41, 49, 53. Also, in line 50 remove “with”; in line 55 change “he” to “The”; in line 65 change “thesis” to “paper” or “study”. Please, read again the entire manuscript and correct it.
- TITLE: You should improve the title. It does not clearly demonstrate the scope of the study. I would suggest “Numerical study on the fluid flow and heat transfer characteristics of Al2O3-water nanofluids in microchannels of different aspect ratio”
- ABSTRACT: In line 13 I suggest to change “surface” to “interface”; in line 15 change “Changing” to “Increasing”, if you think this is the meaning of the sentence.
- INTRODUCTION: The introduction only tries to summarize the relative literature on microchannels. However, since your study focuses on nanofluids, please write, at least, 1 or 2 small paragraphs in order to mention other important factors that affect the thermal conductivity of nanofluids (apart from nanoparticle volume fraction) along with relevant references (Here, I suggest some recent references by proving their link. Of course, you can use also other ones), namely:
- Fundamentals and theory à https://www.sciencedirect.com/science/article/pii/S0370157318303302
https://www.mdpi.com/2076-3417/11/6/2525
https://www.mdpi.com/1996-1944/14/5/1291
- Effect of interfacial nanolayer à
https://www.sciencedirect.com/science/article/pii/S0017931005001250
https://www.tandfonline.com/doi/abs/10.1080/01457632.2019.1692487?journalCode=uhte20
- Effect of aggregations à
https://www.sciencedirect.com/science/article/abs/pii/S0169260720316114?via%3Dihub
- Effect of Brownian motion à
https://www.sciencedirect.com/science/article/pii/S0017931005001328
Mention also some important applications of nanofluids such as:
- heat exchangers by exploiting Al2O3-water nanofluids. See for example:
https://www.sciencedirect.com/science/article/abs/pii/S2451904920303383
https://www.sciencedirect.com/science/article/abs/pii/S2451904920302146?via%3Dihub
- hyperthermia cancer therapy. See for example:
https://www.sciencedirect.com/science/article/abs/pii/S016926071930032X?via%3Dihub
- See for example:
https://www.nature.com/articles/s41598-020-62830-1
- water purification: See for example:
https://www.mdpi.com/2073-4441/11/6/1135
Finally, highlight the innovation of the present study.
- NUMERICAL METHOD AND MODEL DESCRIPTION:
- No information is provided about the numerical methodology, namely finite element/volumes/difference; mesh information; did you developed any in-house numerical model or you use a commercial one? If yes, which one? Please also provide a grid independence study.
- For brevity, you can simply change 3.1 and 3.2 to “Influence of nanoparticle volume fraction” and “Influence of the aspect ratio of microchannels”, respectively.
- Please add in Figs 4-9 the case with a=0% (pure water) as a means of directly observing the impact of the nanofluids on heat transfer. This is very important.
- In line 70 change “known” to “considered”; In line 74 change “equation” to “equations”; In all equations change "f" to "nf", since "f" is usually used for the base fluid in the relative literature; In line 79 remove the bold format.
- CONCLUSION: State again the innovation of your study and highlight its strengths and limitations.
Author Response

(The authors gave the same response as above.)

Round 2
Reviewer 1 Report
In paragraphs 50, 54 reference is made to several studies [9-15] [16-21]. It is considered that it should not include the references and make a short description of what happens in each case study;
- Error analysis is missing!
- Correlation of other authors needs to be shown, and no discussions, no comparisons of the results with the results of other authors.
Please answer questions in Author Response and do not submit a new version of the article that is available on the website.
Author Response
1.The relevant literature has been modified to briefly describe what happened in each case study;
2.Error analysis of the model added in the revised version;
3.Correlation of other authors has been added to the revised version.
Reviewer 3 Report
The authors have responded appropriately to the issues raised by the reviewers. Thus I recommend the paper for publication in its ctual form.
Author Response
Thank you!
Round 3
Reviewer 1 Report
Accept in present form